# 18F-FDG PET-Based Imaging of Myocardial Inflammation Predicts a Functional Outcome Following Transplantation of mESC-Derived Cardiac Induced Cells in a Mouse Model of Myocardial Infarction

**DOI:** 10.3390/cells8121613

**Published:** 2019-12-11

**Authors:** Praveen Vasudevan, Ralf Gaebel, Piet Doering, Paula Mueller, Heiko Lemcke, Jan Stenzel, Tobias Lindner, Jens Kurth, Gustav Steinhoff, Brigitte Vollmar, Bernd Joachim Krause, Hueseyin Ince, Robert David, Cajetan Immanuel Lang

**Affiliations:** 1Department of Cardiac Surgery, Rostock University Medical Center, 18057 Rostock, Germany; praveen.vasudevan@med.uni-rostock.de (P.V.); Ralf.Gaebel@med.uni-rostock.de (R.G.); Paula.Mueller@uni-rostock.de (P.M.); Heiko.Lemcke@med.uni-rostock.de (H.L.); Gustav.Steinhoff@med.uni-rostock.de (G.S.); Robert.David@med.uni-rostock.de (R.D.); 2Department of Life, Light and Matter, University of Rostock, 18059 Rostock, Germany; 3Department of Nuclear Medicine, Rostock University Medical Center, 18057 Rostock, Germany; Piet.doering@uni-rostock.de (P.D.); Jens.Kurth@med.uni-rostock.de (J.K.); Bernd.Krause@med.uni-rostock.de (B.J.K.); 4Rudolf-Zenker-Institute for Experimental Surgery, Rostock University Medical Center, 18057 Rostock, Germany; 5Core Facility Multimodal Small Animal Imaging, Rostock University Medical Center, 18057 Rostock, Germany; Jan.Stenzel@med.uni-rostock.de (J.S.); Tobias.Lindner@med.uni-rostock.de (T.L.); Brigitte.Vollmar@med.uni-rostock.de (B.V.); 6Department of Cardiology, Rostock University Medical Center, 18057 Rostock, Germany; Hueseyin.Ince@med.uni-rostock.de

**Keywords:** inflammation, 18F-FDG PET, cardiomyocytes, cardiac induced cells, cardiac function, non-invasive imaging

## Abstract

Cellular inflammation following acute myocardial infarction has gained increasing importance as a target mechanism for therapeutic approaches. We sought to investigate the effect of syngeneic cardiac induced cells (CiC) on myocardial inflammation using 18F-FDG PET (Positron emission tomography)-based imaging and the resulting effect on cardiac pump function using cardiac magnetic resonance (CMR) imaging in a mouse model of myocardial infarction. Mice underwent permanent left anterior descending coronary artery (LAD) ligation inducing an acute inflammatory response. The therapy group received an intramyocardial injection of 10^6^ CiC into the border zone of the infarction. Five days after myocardial infarction, 18F-FDG PET was performed under anaesthesia with ketamine and xylazine (KX) to image the inflammatory response in the heart. Flow cytometry of the mononuclear cells in the heart was performed to analyze the inflammatory response. The effect of CiC therapy on cardiac function was determined after three weeks by CMR. The 18F-FDG PET imaging of the heart five days after myocardial infarction (MI) revealed high focal tracer accumulation in the border zone of the infarcted myocardium, whereas no difference was observed in the tracer uptake between infarct and remote myocardium. The CiC transplantation induced a shift in 18F-FDG uptake pattern, leading to significantly higher 18F-FDG uptake in the whole heart, as well as the remote area of the heart. Correspondingly, high numbers of CD11^+^ cells could be measured by flow cytometry in this region. The CiC transplantation significantly improved the left ventricular ejection function (LVEF) three weeks after myocardial infarction. The CiC transplantation after myocardial infarction leads to an improvement in pump function through modulation of the cellular inflammatory response five days after myocardial infarction. By combining CiC transplantation and the cardiac glucose uptake suppression protocol with KX in a mouse model, we show for the first time, that imaging of cellular inflammation after myocardial infarction using 18F-FDG PET can be used as an early prognostic tool for assessing the efficacy of cardiac stem cell therapies.

## 1. Introduction

Inflammatory activity of the innate immune system following myocardial infarction substantially influences remodeling after myocardial infarction (MI) and the evolution of heart failure [1]. Therefore, anti-inflammatory therapies have been under intense investigation since the early 1970s. Yet, none of the clinical trials testing pharmacological strategies aiming at unspecific reduction of myocardial inflammation has proven successful [2]. The main reason for these disappointing findings is the complexity of the well-orchestrated activity of different leukocyte populations. The amplitude and duration of inflammation and the timely and spatial resolution within the heart define the quality of the scar following MI and the amount of tissue loss rather than the mere extent of inflammation [1]. Hence, the concept of suppressing inflammation has changed to a further elaborated concept of “modulating cellular inflammation”. The balance of pro- and anti-inflammatory immune cells and their recruiting mechanisms are discussed as both therapeutic and prognostic targets [1]. Further understanding of this intricate cellular immune response to myocardial ischemia in a translational setting requires non-invasive molecular imaging tools. The 18F-FDG PET (Positron emission tomography) using specific protocols for suppressing glucose uptake in cardiomyocytes has recently been introduced to detect cellular inflammation following MI in both patients and C57BL/6 mice [1,3,4,5] 

Protocols to suppress physiological myocardial 18F-FDG uptake require low-carbohydrate diet the day before imaging followed by a 12 h fasting period and intravenous application of heparin before the actual scan [3]. In mice, myocardial glucose metabolism can be effectively suppressed by replacing the commonly used anaesthetic isofluorane for a ketamine/xylazine-based protocol [1].

However, to our knowledge, no studies have used these protocols for measuring therapeutic effects of therapies aiming at improving myocardial healing following MI. Therefore, we implanted syngeneic cardiac induced cells committed to the cardiac lineage in order to improve post-MI cardiac function in 129Sv mice as measured my cardiac magnetic resonance (CMR). Cellular inflammation was detected by 18F-FDG PET in vivo and by post-mortem flow cytometry in both infarcted and remote myocardium. 

We hypothesize that PET-based imaging and quantification of cellular inflammation can be used as a molecular imaging tool for both quantification and spatial distribution of monocytes and as an early prognostic tool to predict the effect of cardiac stem cell therapies modulating post-MI inflammation.

## 2. Materials and Methods

### 2.1. Stem Cell Culture and Cardiovascular Differentiation

W4 murine embryonic stem cells (mESCs), originally isolated from the 129S6 mouse strain [6], were grown according to standard protocols as described previously [7]. In brief, cells were cultured in DMEM supplemented with 15% FBS Superior (Biochrom AG, Berlin, Germany), 1% Cell Shield^®^ (Minerva Biolabs GmbH, Berlin, Germany), 100 µM non-essential amino acids, 1000 U/mL leukemia inhibitory factor (Phoenix Europe GmbH, Mannheim, Germany) and 100 µM β-mercaptoethanol (Sigma-Aldrich GmbH, Steinheim, Germany) at 37 °C, 5% CO_2_, and 20% O_2_. For cardiovascular differentiation we used cardiogenic differentiation medium, containing IBM (Iscove’s Basal Medium, Biochrom AG) supplemented with 10% FBS Superior, 1% Cell Shield^®^, 100 µM non-essential amino acids, 450 µm 1-thioglycerol (Sigma-Aldrich GmbH), and 213 µg/mL ascorbic acid (Sigma-Aldrich GmbH), as described previously [7,8]. Cardiovascular differentiation was initiated by hanging-drop culture for two days at 37 °C, 5% CO_2_, and 20% O_2_. 400 cells per drop were plated on the cover of a square petri dish and grown for 2 days to start formation of embryoid bodies (EB). Afterwards, EB were grown additional four days in suspension culture [9] and then harvested for transplantation and PCR analysis. These cells will be referred to as cardiac induced cells (CiC). They were cultured till day 30 for beating foci analysis. 

### 2.2. Animal Model

The present study was approved by the federal animal care committee of the Landesamt für Landwirtschaft, Lebensmittelsicherheit und Fischerei Mecklenburg-Vorpommern (LALLF, Germany) (registration no. LALLF M-V/TSD/7221.3-1.1-054/15). The 129S6/SvEvTac were bred in the animal facility of the Rostock University Medical Center and maintained in specified pathogen-free conditions. The mice had access to water and standard laboratory chow ad libitum and received humane care according to the German legislation on protection of animals and the Guide for the Care and Use of Laboratory Animals (NIH publication 86–23, revised 1985), and all efforts were made to minimize suffering. Mice were anaesthetized with pentobarbital (50 mg/kg, intraperitoneal). Following thoracotomy, the left anterior descending coronary artery (LAD) was ligated to induce acute myocardial infarction. The MI group received an intramyocardial injection of 10 µL PBS mixed with 10 µL Growth Factor Reduced Matrigel™ Matrix (Corning, Berlin, Germany). The MI induction plus cell transplantation (MIC) group received a suspension of 1 × 10^6^ syngeneic in PBS (10µL) mixed with 10 µL Growth Factor Reduced Matrigel™ Matrix. Injections of 4 × 5 µL were given along the infarct border. The injection site was controlled visually at the time of transplantation [10].

Healthy mice (n = 33) were divided into 7 groups for 18F-FDG PET imaging 5 days following MI using different protocols for anaesthesia: (1)no intervention, isofluorane (n = 2)(2)no intervention, ketamine/xylazine (n = 2)(3)MI group, isofluorane (n = 4)(4)MI group, ketamine/xylazine (n = 6)(5)MIC group, ketamine/xylazine (n = 7)

Cardiac function was assessed in separate groups three weeks following MI by cardiac MRI:(6)MI only (n = 6)(7)MIC (n = 6)

Flow cytometric analysis was performed in separate groups (MI, n = 5 and MIC, n = 5).

### 2.3. qRT-PCR

RNA was isolated from the cells using the NucleoSpin^®^ RNA isolation kit (Macherey-Nagel, Dueren, Germany). First strand cDNA was then synthesized using the cDNA synthesis kit (Thermo Fisher Scientific, Waltham, MA, USA) according to the manufacturer’s instructions. The qPCR reaction was then carried out with the Taqman^®^ Universal PCR Master Mix (Thermo Fisher Scientific) and performed on a StepOnePlus Real-Time PCR system (Applied Biosystems, Foster city, CA, USA). Primers of the following target genes: *Pou5f1* (Mm00658129_gH), *cTnnt2* (Mm01290256_m1), *MesP1* (Mm00801883_m1), and *Nkx2.5* (Mm01309813_s1) were purchased from Thermo Fisher Scientific. Gene expression values of the target genes at day 6 were then normalized to the housekeeping gene *Hprt* (Mm00446968_m1; Thermo Fisher Scientific) and compared relative to the expression values at day 0 using the ∆∆Ct method for relative quantifications.

### 2.4. Beating Foci Analysis

The number of beating foci per EB was analyzed from day 7 to day 30 of differentiation. The EB were observed under a microscope (Carl Zeiss, Oberkochen, Germany) and the beating foci per each EB were then visually analyzed using the ZEN2011 software (Carl Zeiss).

### 2.5. Flow Cytometry

Single cell cardiac monocyte suspensions were prepared for flow cytometry, as previously described [11] Briefly, the remote and infarct tissue of the heart was dissected and enzymatically digested separately in HBSS with Ca^2+^ and Mg^2+^(450 U/mL collagenase type I, 125 U/mL collagenase type XI, 120 U/mL DNase I, 60 U/mL hyaluronidase, all Sigma-Aldrich) for 30 min at 37 °C. The digested samples were then passed through a 100 µm filter and centrifuged to enrich for mononuclear cells. Red blood cells were then lysed using erythrocytes lysis buffer (eBioscience, San Diego, CA, USA) and the digest was then washed and suspended in MACS^®^ buffer (PBS, 2 mM EDTA, 0.5% BSA). Samples were then labeled using Zombie Aqua dye (BioLegend, San Diego, CA, USA.), washed, resuspended in MACS buffer containing FCR Block (Miltenyi Biotec GmbH, Bergisch Gladbach, Germany), and stained (see Table 1 for antibody list). Stained samples were then analyzed on a BD FACS LSR II^®^ running BD FACS Diva software (version 6.1.2, Franklin Lakes, NJ, USA). The various immune cell populations in the heart tissue were then assessed, as described in Figure 1.

### 2.6. PET Imaging

For the PET study, mice of the groups 1,3, 6, and 7 were anaesthetized by inhalation of isoflurane (4% for induction and 1% to 2.5% maintenance during preparation and scanning), whereas mice of the groups 2,4, and 5 were anaesthetized by i.p. injection of ketamine/xylazine (ketamine 84 mg/kg and xylazine 11.2 mg/kg) 20 min before tracer application. All PET/CT scans were performed on a small animal PET/CT scanner (Inveon MM-PET/CT, Siemens Medical Solutions, Knoxville, TN, USA) [12] according to a standard protocol: Mice were injected intravenously with a dose of approximately 10 MBq 18F-FDG via a custom-made micro catheter placed in a tail vein. After an uptake period of 60 min, mice were imaged in prone position for 20 min. During the imaging session, respiration of the mice was controled and core body temperature was constantly kept at 38 °C via a heating pad. For attenuation correction and anatomical reference, whole body CT scans were acquired. CT images were reconstructed with a Feldkamp algorithm. The PET images were reconstructed with the three-dimensional (3D) iterative ordered subset expectation maximization reconstruction algorithm (3D-OSEM/OP-MAP) with the following parameters: 4 iterations (OSEM), 32 iterations (MAP), 1.7 mm target resolution, and 128 × 128 matrix size. Reconstruction included corrections for random coincidences, dead time, attenuation, scatter, and decay.

### 2.7. PET Image Analysis 

Image analyses were performed using an Inveon Research Workplace (Siemens, Knoxville, TN, USA), as described previously [13]. PET (Positron emission tomography) and CT (Computerized tomography) images were fused by the use of an automated volumetric fusion algorithm and then verified by an experienced reader for perfect alignment. Consecutively, standardized representative volumes of interest (VOI) were manually placed in the remote area and in the infarcted region as well as the whole heart guided by anatomical landmarks, as described in detail in Figure 2. Correct VOI positioning was visually verified in axial, coronal, and sagittal projection. 

Carimas 2 software (Turku PET Centre, Turku, Finland) was used for generating polar maps of the left ventricle according to the manual provided by the developer. Results are presented using 17-segmental standardized myocardial segmentation.

### 2.8. Cardiac Magnetic Resonance Imaging

Cardiac magnetic resonance (CMR) measurements were performed on a 7 Tesla small animal MRI system (BioSpec 70/30, maximum gradient strength 440 mT/m, Bruker BioSpin Gmbh, Ettlingen, Germany) equipped with a 1H transmit volume coil (86 mm, volume resonator) and a 2-by-2 receive-only surface coil array (both Bruker BioSpin GmbH). After induction of anaesthesia using 2% to 3.5% isoflurane in oxygen, animals were placed in a supine position on a dedicated mouse bed and surface coil was placed on the chest of the mice. Respiration rate and body temperature were monitored using an MR-compatible small animal monitoring and gating system (Model 1030, SA Instruments, Inc., Stony Brook, NY, USA), and stable body temperature was maintained by a warm water heating. Anaesthesia was maintained during the experiment with isoflurane oxygen (1.5% to 2%) to achieve a respiration rate of about 35 to 55 breaths.

After planning sequences, for the short axes view final images of the left ventricular ejection fraction (LVEF) measurements were acquired using a IntraGate gradient-echo cine sequences (IntraGate Cine-FLASH) in six short-axis planes completely covering the left ventricle. Acquisition parameters included: echo time (TE) 2.38 ms, repetition time (TR) 5.89 ms, flip angle 15°, 14 frames per cardiac cycle, oversampling 140, averages 1, field of view (FOV) 29.4 × 25.2 mm, matrix size 211 × 180, resolution in-plane 0.14 × 0.14 mm, slice thickness 1 mm, and scan time per slice 2 min.

### 2.9. Cardiac Magnetic Resonance Analysis

LVEF was assessed from the cine sequences using the freely available software Segment v2.0 R5165 (Medviso, Lund, Sweden) (http://segment.heiberg.se) [14]. The left ventricular (LV) endocardium in these slices was manually segmented to exclude the papillary muscles. The volumes of these segments were then integrated along the six planes of the LV and LVEF was then calculated from their summated end systolic and end diastolic volumes.

### 2.10. Statistics

All data are presented as mean values ± standard deviation (SD). Flow cytometric data and qPCR values are presented as mean values ± standard error of mean. Student’s *t*-test was used for statistical analysis of parametric data and the Mann–Whitney test was used for nonparametric analysis of flow cytometric data. Values of *p* < 0.05 were considered statistically significant.

## 3. Results

### 3.1. Cardiac Induced Cells Show Increased Cardiac Markers and Beating Activity During Differentiation

In order to ascertain the differentiation status of the cells, we examined the expression of various markers at the beginning and at day six of differentiation (Figure 3A). We observed the expression of early cardiac markers such as *MesP1*, *Nkx2.5*, and cardiac troponin (*cTnnt2*). They were increased several fold at day six, whereas the expression of the pluripotency marker, *Pou5f1* (also known as *Oct4*), was strongly decreased. This indicates the cardiac lineage commitment of the cells. We also investigated the beating activity of the cells along the differentiation process to determine their functional capability (Figure 3B). In accordance with the strong induction with the induction of the cardiac markers, the cells were robustly beating and the number of beating foci per EB consistently increases over time till day 30 of differentiation. Therefore, these cardiac-induced cells have committed to the cardiac lineage with the potential to form beating cells in the host after transplantation.

### 3.2. Impact of Anaesthesia on the FDG-Uptake Pattern in the Infarcted Heart

First, we examined if the myocardial glucose uptake suppression protocol established by Thackeray et al. [5] for C57BL/6 J mice can be applied in the mouse strain 129Sv used in our study on healthy mice (Figure 4).

Secondly, 18F-FDG PET was performed five days after permanent LAD ligation. Under anaesthesia with isofluorane, the highest tracer accumulation was detected in the viable myocardium, whereas ketamine/xylazine (KX) led to accentuated tracer accumulation within the border zone (Figure 5).

The 18F-FDG uptake was significantly reduced by the use of ketamine/xylazine as compared with isofluorane for anaesthesia in the whole heart (5.2 ± 0.7% ID/g vs. 46.1 ± 11.2% ID/g; *p =* 0.02) and both remote (4.1 ± 0.6% ID/g vs. 79.3 ± 22.7% ID/g; *p* < 0.0001) and infarcted myocardium (4.35 ± 0.4% ID/g vs. 11.6 ± 6.0% ID/g, *p =* 0.002) (Figure 6).

With KX, there was no difference in tracer uptake between remote and infarcted myocardium (4.1 ± 0.6% ID/g vs. 4.4 ± 0.4% ID/g; *p =* 0.5). In contrast isoflurane lead to significantly higher 18F-FDG uptake in the remote as compared with the infarcted myocardium (79.3 ± 22.7% ID/g vs. 11.6 ± 6.0% ID/g; *p =* 0.001).

### 3.3. FDG-Uptake Pattern is Fundamentally Changed by Cell Transplantation

We then compared 18F-FDG uptake patterns in untreated to animals treated with CiC therapy (MIC) using the ketamine/xylazine protocol (Figure 7 and Figure 8). Transplantation of embryoid bodies containing 10^6^ syngeneic cardiac induced cells following acute myocardial infarction led to an increase in 18F-FDG uptake in the remote myocardium as compared with the MI group (10.7 ± 4.3% ID/g vs. 4.1 ± 0.6% ID/g; p = 0.003) (Figure 9). Interestingly, tracer accumulation in the center of the infarcted area was not altered by cell therapy (4.3 ± 1.4% ID/g in MIC vs. 4.4 ± 0.4% ID/g in MI; *p =* 0.9). Furthermore 18F-FDG uptake in the whole heart was significantly increased in the MIC group (8.0 ± 2.9% ID/g vs. 5.2 ± 1.1% ID/g; *p* < 0.05).

### 3.4. Improvement of Cardiac Function Through Cell Therapy Assessed by CMR

In order to assess the value of 18F-FDG-based imaging of cellular inflammation post-MI as an early predictor of functional outcome following cardiac cell therapy, cine CMR was performed three weeks after LAD ligation in separate groups. LVEF was significantly reduced by LAD ligation as compared with healthy animals (24.2 ± 4.1% vs. 59.3 ± 3.7%; *p* < 0.001). Cell therapy led to a significant increase of LVEF (31.7 ± 3.5% vs. 24.2 ± 4.1%; *p =* 0.007) (Figure 10). From this we conclude, that the change in the 18F-FDG uptake pattern, as described above, is a valuable early predictor of therapeutic efficacy.

### 3.5. Modulation of the Immune Response Through Cell Therapy Assessed by Flow Cytometry

In order to identify the immune response within the myocardium, we analyzed the mononuclear cell (MNC) suspension isolated from the remote and infarct area of the hearts of 129Sv mice five days after MI or MIC (Figure 1).

We observed a significantly higher percentage of NK cells (CD45^+^/CD11b^−^/CD11c^+^/NK1.1^+^) in the infarct area (0.82 ± 0.04 vs. 0.56 ± 0.03; *p =* 0.01) as compared with the remote area in MI mice. Cell therapy influences the various immune subpopulations differently in infarct and remote areas of the heart. On the one hand, cell therapy led to a significantly higher percentage of CD11b^+^ myeloid cells (Figure 11A) in the remote area (49.28 ± 3.92 vs. 31.17 ± 2.72; *p =* 0.01) of MIC hearts as compared with MI hearts. This agrees well with the increased 18F-FDG uptake observed in the remote area of MIC mice (Figure 9), which has been attributed mainly to CD11b^+^ cells [1]. On the other hand, cell therapy led to a decrease in the percentage of NK cells (Figure 11A) in the infarct area (0.41 ± 0.05 vs. 0.82 ± 0.04; *p =* 0.007) when compared to MI hearts.

Interestingly, we also observed various differences in various CD11b^+^ subpopulations based on their relative CCR2 and MHC-II expression between the MI and MIC groups (Figure 11B,C).

In the remote area, cell therapy led to an increase in the percentage of fetal liver HSC-derived resident macrophages (4.68 ± 1.03 vs. 1.29 ± 0.32; *p =* 0.05) and an increase in the percentage of monocyte-derived macrophages (50.48 ± 3.24 vs. 24.35 ± 4.54; *p =* 0.007), and their relative contribution to the percentage of M1 (Ly6C^hi^) (58.48 ± 5.41 vs. 28.11 ± 7.72; *p =* 0.01) and M2 (Ly6C^lo^) (47.28 ± 3.95 vs. 23.39 ± 3.38%; *p =* 0.007) cells along with a subsequent decrease in the percentage of monocytes (24.33 ± 5.61 vs. 44.36 ± 5.62; *p =* 0.05) and their relative contribution to the percentage of M1 (Ly6C^hi^) (33.76 ± 6.9 vs. 67.85 ± 7.22; *p =* 0.03) and M2 (Ly6C^lo^) (13.25 ± 1.74 vs. 32.71 ± 3.56; *p =* 0.007) cells as compared with MI.

In the infarct area, we observed an increase in the percentage of fetal liver HSC-derived resident macrophages (4.68 ± 1.03 vs. 1.29 ± 0.32; *p =* 0.05) as well as monocyte-derived macrophages (57.46 ± 4.37 vs. 38.89 ± 8.13) and their relative contribution to the percentage of M2 (Ly6C^lo^) (51.44 ± 2.47 vs. 33.25 ± 6.6%; *p =* 0.03) cells with a subsequent decrease in the percentage of monocytes (18.88 ± 5.99 vs. 32.25 ± 6.88; *p =* 0.05) and their relative contribution to the percentage of M1 (Ly6C^hi^) (27.6 ± 7.59 vs. 47.4 ± 10.04; *p =* 0.05) and M2 (Ly6C^lo^) (8.39 ± 1.17 vs. 25.85 ± 6.15; *p =* 0.007) along with a decrease in the percentage of yolk sac-derived resident macrophages (20.16 ± 2.43 vs. 27.95 ± 2.15; *p =* 0.05) as compared with the MI group. 

Therefore, CiC transplantation in the heart increases the CD11b^+^ cells in the remote area of the heart while favoring an increase in the MHC-II^hi^ subset of CD11b^+^ cells (monocyte derived macrophages and fetal liver HSC-derived resident macrophages) in both the infarct and remote areas.

## 4. Discussion

Translational cardiovascular research has evolved at an incredible pace within the last decade and resulted in significantly higher survival rates after acute myocardial infarction. In contrast to highly efficient approaches based on early revascularization and pharmacological prevention of adverse ventricular remodeling, replacement of irreversibly lost cardiomyocytes has not been achieved yet, despite huge efforts in the field of regenerative medicine.

Recently, cellular inflammation following ischemic myocardial injury has been identified as a key player in the process of myocardial healing. Thereby, the local distribution patterns of specific monocyte and macrophage subpopulations have been proposed to determine the quality of myocardial healing [15,16].The vast majority of data has been obtained from rodent studies because their hearts can easily be excised for in vitro experiments such as flow cytometric measurements of single cells suspensions. Most researchers focus on the invasion of two distinct macrophage subpopulations, usually referred to as M1 and M2 macrophages. The early M1 macrophage subset is attracted to the site of myocardial injury via CCL2, expressing pro-inflammatory mediators and proteases for degradation of infarcted tissue. Subsequently, M2 macrophages are recruited via CX3CL1 for mediating synthesis of extracellular matrix and angiogenesis [3]. However, the M1/M2 classification does not adequately explain the complete spectrum of macrophage phenotypes. Recently, macrophages which do not express CCR2 in the neonatal heart have been shown to regenerate the infarcted tissue [15] while the adult heart involves CCR2^+^ monocyte-derived macrophages also taking part in the remodeling process [17].

Preclinical studies have examined the effect of therapeutic applications, such as stem cell injections, to modify the qualitative and quantitative composition of the post infarct cellular immune response [18,19,20]. This modulation of the innate cellular immune response resulted in improved cardiac pump function, reduction of scar size, and adverse remodeling [18]. Findings from a meta-analysis by our group have ascribed high therapeutic potential to cardiovascular cell preparations [21]. Therefore, we sought to transplant ESC-derived cardiac induced cells to improve myocardial healing and also investigate whether they influence the ischemic cellular immune response.

The above-mentioned experiments are based on post-mortem analyses such as flow cytometry and immunohistochemistry of excised organs, and thus have been restricted to preclinical research. However, clinical translation requires methods that allow in-vivo visualization and quantification of inflammatory cells. Lee et al. established 18F-FDG PET for imaging the inflammatory cell activity in the heart, based on suppressing glucose metabolism in myocytes [1]. Interestingly, anaesthesia with ketamine/xylazine is both sufficient and highly effective in reducing glucose uptake in cardiomyocytes, hence, enabling visualization of inflammatory activity in the heart [1]. According to Lee et al., 18F-FDG uptake at the site of myocardial inflammation related to the content of local CD11b^+^ monocyte/macrophage concentration in C57BL6 mice [1], at day 5 after MI induction.

On the basis of these findings by other groups, we hypothesized the following:(A)Distribution pattern of CD11b^+^ myeloid cells at day five after MI induction can be modified by intramyocardial transplantation of CiC;(B)This change can be visualized and quantified by 18F-FDG PET using ketamine/xylazine anaesthesia;(C)The specific 18F-FDG uptake pattern correlates with functional outcome as measured by cardiac magnetic resonance imaging.

We were able to replicate the glucose uptake suppression protocol based on anaesthesia with KX [5] in 129sv mice and achieved an almost 88% reduction in the glucose uptake in the whole myocardium as compared with isoflurane anaesthesia. We did not find any significant difference in the 18F-FDG uptake levels between the remote and infarcted myocardium using KX anaesthesia, which is in line with findings from Thackeray et al. [4,5]. Furthermore, intense focal tracer uptake was localized in the border zone. Similarly, Lee et al. also reported accentuated FDG uptake in the border zone corresponding to high numbers of infiltrating CD11b^+^ cells [1]. Despite visual assessment of this high focal tracer accumulation, quantitative analysis in this region was only performed by Thackeray et al. However, their VOI positioning strategy remains unclear from the manuscript. This might be due to a certain overlap of the border zone to the adjacent regions which makes demarcation of the border zone difficult. In order to produce comparable data in this aspect, VOI positioning strategies should be provided in future studies.

Interestingly, transplantation of 10^6^ cardiac induced cells after MI significantly increased FDG uptake in both the whole heart and the remote myocardium, but not in the infarct region. Moreover, CiC transplantation led to a 7% improvement in LVEF three weeks after MI. The magnitude of LVEF improvement is well in line with previous reports [21]. Therefore, we conclude that the change in the myocardial 18F-FDG uptake pattern represents a valid tool for early PET-based in vivo prediction of myocardial healing post MI.

A deeper investigation of the immune response in the heart five days after MI using flow cytometry revealed an increase in the percentage of CD11b^+^ cells and a shift towards increased monocyte-derived macrophages in the remote area of cell-treated mice. We also observed a reduction in the yolk sac-derived resident macrophages in the infarct area of cell-treated mice. We did not however observe a shift in the M1/M2 polarization phenotype, as observed in previous studies involving MSCs [18]. It is interesting to note that increased monocyte-derived macrophages and reduced yolk sac-derived macrophages in the heart have been previously attributed to adverse cardiac remodeling and scar formation [17]. However, these studies did not involve the transplantation of cells into the infarcted heart and it is well known that the gene expression profile and the characteristics of the immune cells are affected based on their location in the myocardium and the cells they interact with [22].

However, unbiased expression profiling of these cells over various time points and under the influence of CiC has yet to be carried out. The influence of CiC transplantation on the immune response in the heart has not been studied until now and our work suggests possible new mechanisms and targets for improving the efficiency of CiC.

The 18F-FDG-based imaging strategies for myocardial inflammation are highly attractive and have been applied in a clinical setting despite some limitations, which have yet to be overcome [3]. Both cardiomyocytes [4] and infiltrating inflammatory cells [23] in the setting of acute myocardial injury possess high levels of glucose metabolism. Metabolism of healthy cardiomyocytes is mainly based on fatty acid oxidation, whereas ischemia triggers increased anaerobic glycolysis, which requires much higher rates of glucose [24]. This might hamper direct correlation of focal 18F-FDG uptake and high concentration of CD11b^+^ cells in the infarcted region. The use of KX for anaesthesia has been shown to reduce serum insulin levels in rodents, thus, preventing translocation of GLUT4 to membranes of cardiomyocytes and reducing 18F-FDG uptake [25]. In contrast, leucocyte glucose influx in a setting of acute inflammation depends more on GLUT1 and GLUT3, which are expressed and translocated independently of insulin [26]. Whereas, this KX protocol is well suited for preclinical research, different strategies are used for suppression of myocardial glucose uptake in patients including prolonged fasting, dietary modifications, and heparin loading before imaging [27]. This might hamper straightforward translation of imaging protocols established in rodents to clinical application.

Furthermore, the transplantation of CiC adds more complexity to its use in imaging the myocardium. It becomes difficult to attribute the observed 18F-FDG uptake pattern to the inflammatory cells alone since the ability of CiC to alter the cardiac metabolism is poorly understood. It is, therefore, difficult to delineate the relative contribution of 18F-FDG uptake between the host cardiomyocytes, immune cells, and the CiC. The heterogeneity of the inflammatory response and the relative 18F-FDG uptake between the different immune cell populations is also clearly not understood. Further research is necessary to understand metabolic changes in the different cells involved in healing myocardium.

Nevertheless, this is the first study to investigate 18F-FDG-based PET imaging of inflammation as an early indicator for assessing long term therapeutic efficiency in a rodent model of acute myocardial infarction. Furthermore, the current work illustrates the benefit of CiC transplantation to improved cardiac function after MI possibly through its beneficial modulation of the innate inflammatory response. Using such non-invasive techniques in the field of translational research will foster a better understanding of the inflammatory response and how its modulation could lead to altered cardiac function. It is also a valuable tool for monitoring various immune modulation and cell therapies and broadens the horizon for understanding the mechanisms of these therapeutic strategies.

## Figures and Tables

**Figure 1 cells-08-01613-f001:**
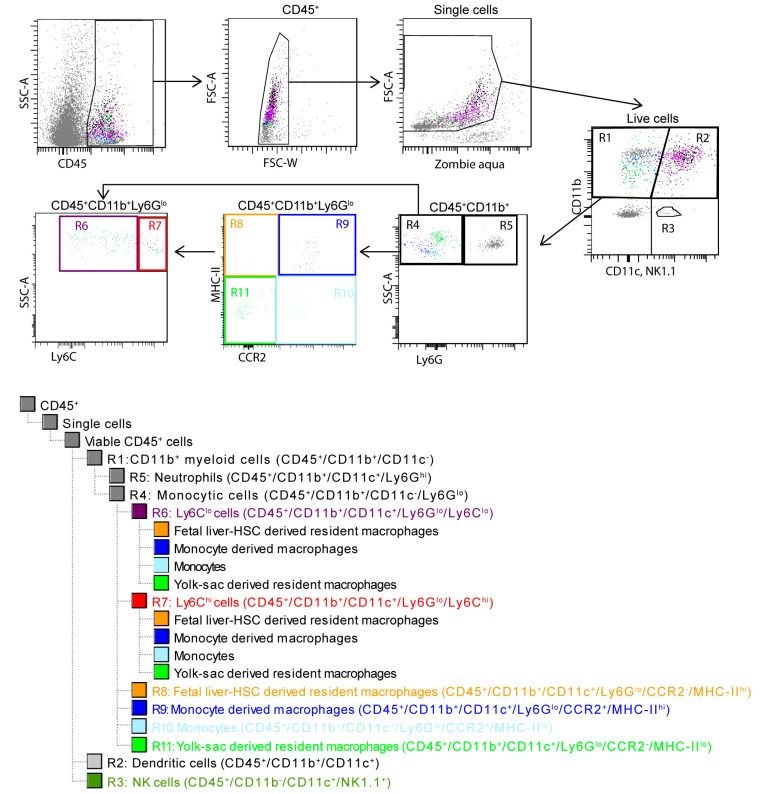
Gating strategy for identifying the different immune populations in the heart. Mononuclear cells expressing CD45 were gated and doublets (FSC-W vs. FSC-A) were excluded. Dead cells were excluded by Zombie aqua. The live single CD45^+^ cells were then grouped into R1, CD11b+ myeloid cells (CD45^+^/CD11b^+^/CD11c^−^); R2, dendritic cells (CD45^+^/CD11b^+^/CD11c^+^); and R3, NK cells (CD45^+^/CD11b^−^/CD11c^−^/NK1.1^+^) based on their relative expression of CD11b and CD11c. R5, neutrophils (CD45^+^/CD11b^+^/CD11c^-^/Ly6G^hi^) were then excluded from R1 based on their Ly6G expression. The remaining R4 monocytic cells were then further characterized into R6, Ly6C^hi^ or commonly known as M1 cells (CD45^+^/CD11b^+^/CD11c^−^/Ly6G^lo^/Ly6C^hi^); R7, Ly6C^lo^ or commonly known as M2 cells (CD45^+^/CD11b^+^/CD11c^−^/Ly6G^lo^/Ly6C^lo^) based on their Ly6C expression; and into R8, fetal liver HSC-derived resident macrophages (CD45^+^/CD11b^+^/CD11c^-^/Ly6G^lo^/CCR2^−^/MHC-II^hi^); R9, monocyte derived macrophages (CD45^+^/CD11b^+^/CD11c^-^/Ly6G^lo^/CCR2^+^/MHC-II^hi^); R10, monocytes (CD45^+^/CD11b^+^/CD11c^−^/Ly6G^lo^/CCR2^+^/MHC-II^lo^); and R11, yolk sac-derived resident macrophages (CD45^+^/CD11b^+^/CD11c^−^/Ly6G^lo^/CCR2^−^/MHC-II^lo^) based on their CCR2 and MHC-II expression. These CCR2 and MHC-II gated populations were then back gated on R6 and R7 and their relative contribution to the M1 (Ly6C^hi^) and M2 (Ly6C^lo^) cells was assessed.

**Figure 2 cells-08-01613-f002:**
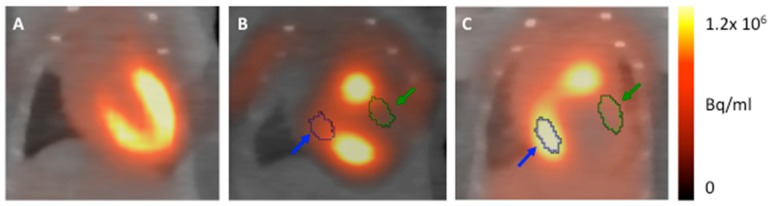
Representative images visualizing our volumes of interest (VOI) positioning strategy: (**A**) Myocardium of healthy mice anaesthetized with isofluorane can be clearly delineated and served as a reference for VOI positioning. (**B** and **C**) Anaesthesia with ketamine/xylazine. A VOI of 5 µL was positioned in both anterolateral wall (infarct area, green) and remote area (inferobasal, blue). Thereby both anatomical landmarks form the CT scan and the image of healthy myocardium (**A**) was used.

**Figure 3 cells-08-01613-f003:**
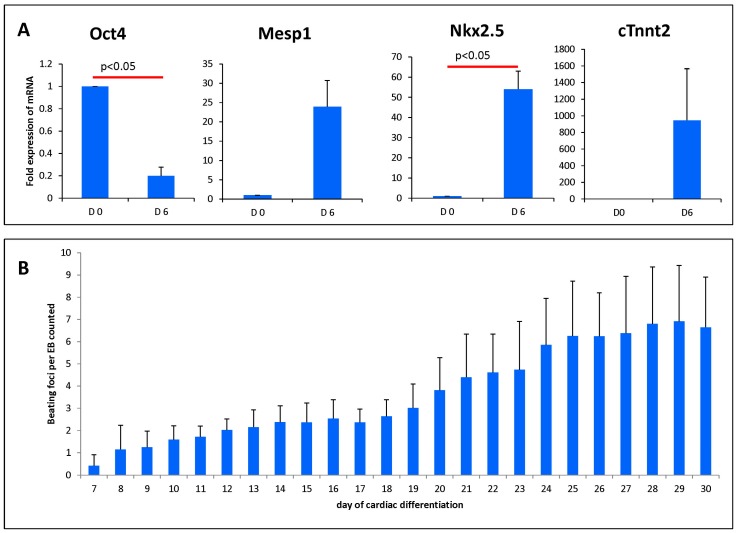
Characterization of cardiac-induced cells: (**A**) Analysis of mRNA expression after six days of cardiac differentiation shows a decrease of the pluripotency marker *Oct4*, whereas cardiac markers are upregulated. The mRNA expression was normalized to the expression of the house keeping gene *hprt*. Values are expressed as fold increase as compared with day 0. Values are presented as mean ± SEM. (**B**) Starting at day 7 of the cardiac differentiation protocol, beating foci per embryoid bodies (EB) were counted until the end of differentiation (n = 3). Values are presented as mean ± SD. *p*-value was calculated using the student *t*-test.

**Figure 4 cells-08-01613-f004:**
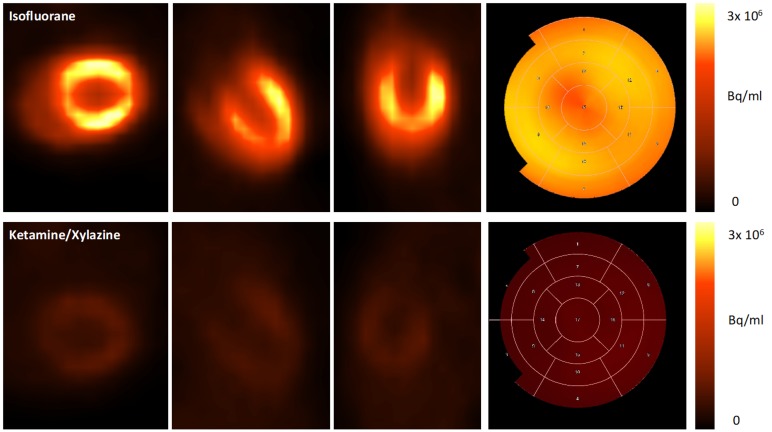
Representative images of healthy mice anaesthetized with isofluorane (upper row) and ketamine/xylazine (lower row). Uptake of 18F-FDG is effectively suppressed by the use of ketamine/xylazine as can be seen in axial, coronal, and sagittal planes, as well as the polar maps.

**Figure 5 cells-08-01613-f005:**
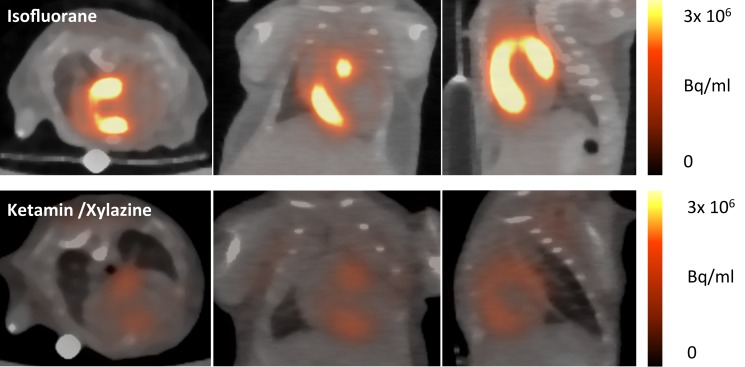
Sample axial, coronal, and sagittal myocardial 18F-FDG images at five days post-surgical myocardial infarction (MI) induction. Isofluorane leads to high tracer accumulation in the healthy myocardium, whereas the infarcted area can be clearly identified as an area of low glucose metabolism (upper row). When using ketamine/xylazine, the most intense tracer accumulation is detected in the area of the border zone, whereas 18F-FDG uptake in healthy myocardium remains suppressed.

**Figure 6 cells-08-01613-f006:**
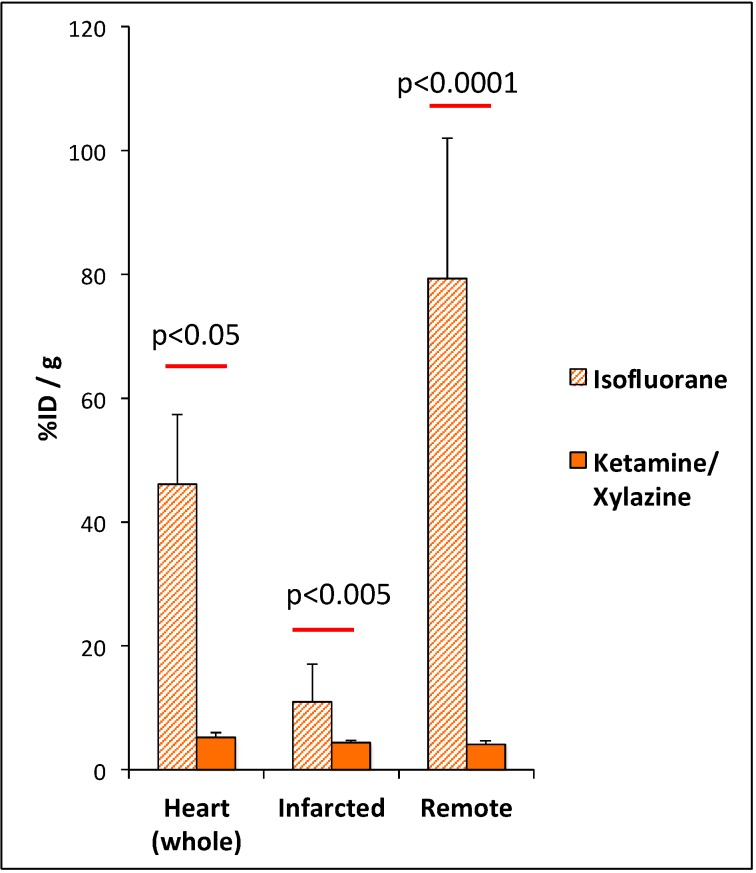
Regional quantitative analysis of 18F-FDG uptake five days after myocardial infarction in the whole heart, the infarct and the remote myocardium. Mice were anaesthetized with isofluorane or ketamine/xylazine, respectively. Values are presented as mean ± SD. *p*-value was calculated using the student *t*-test.

**Figure 7 cells-08-01613-f007:**
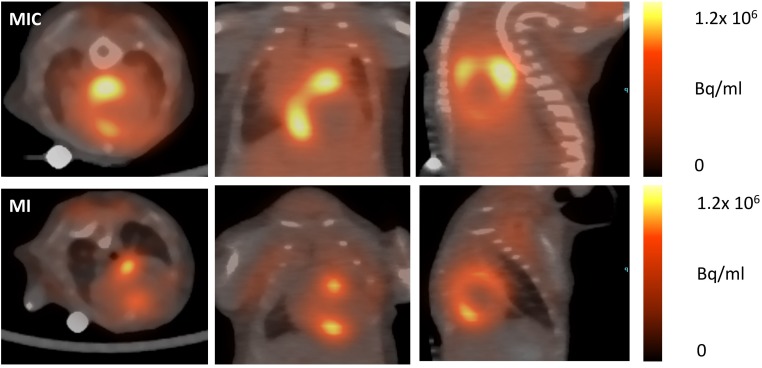
Sample axial, coronal, and sagittal myocardial 18F-FDG images at five days post-surgical MI induction and MI induction plus cell transplantation (MIC). Mice were anaesthetized with ketamine/xylazine. The MI group showed intense tracer accumulation in the border zone, whereas in cell-treated animals the highest tracer accumulation was found in the remote area.

**Figure 8 cells-08-01613-f008:**
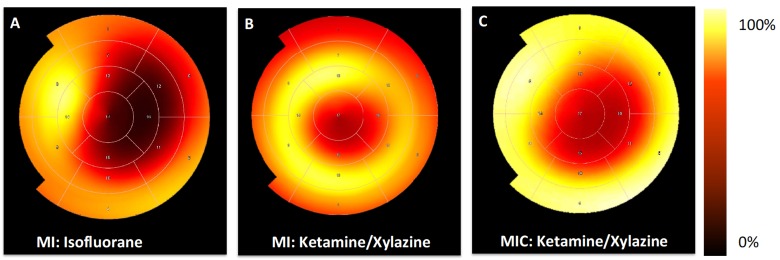
Representative 17-segment tomographic polar maps visualizing distinctive differences of FDG-distribution in the left ventricle between the groups. The apex is in the middle and the anterior wall at top, the inferior wall at bottom, the septum in left and the lateral wall in the right. High tracer uptake is visualized by the yellow colors, lower upake by red and black. (**A**) MI group anaesthetized with isofluorane, the infarct region shows low FDG uptake as compared with remote myocardium; (**B**) MI group anaesthetized with ketamine/xylazine, the most intense FDG uptake came in the border zone; and (**C**) MI + cardiac induced cells (CiC) group anaesthetized with ketamine/xylazine shows a change of the FDG uptake to a pattern, which is similar to (A) with low FDG accumulation within the infarct region and highest uptake in the remote area.

**Figure 9 cells-08-01613-f009:**
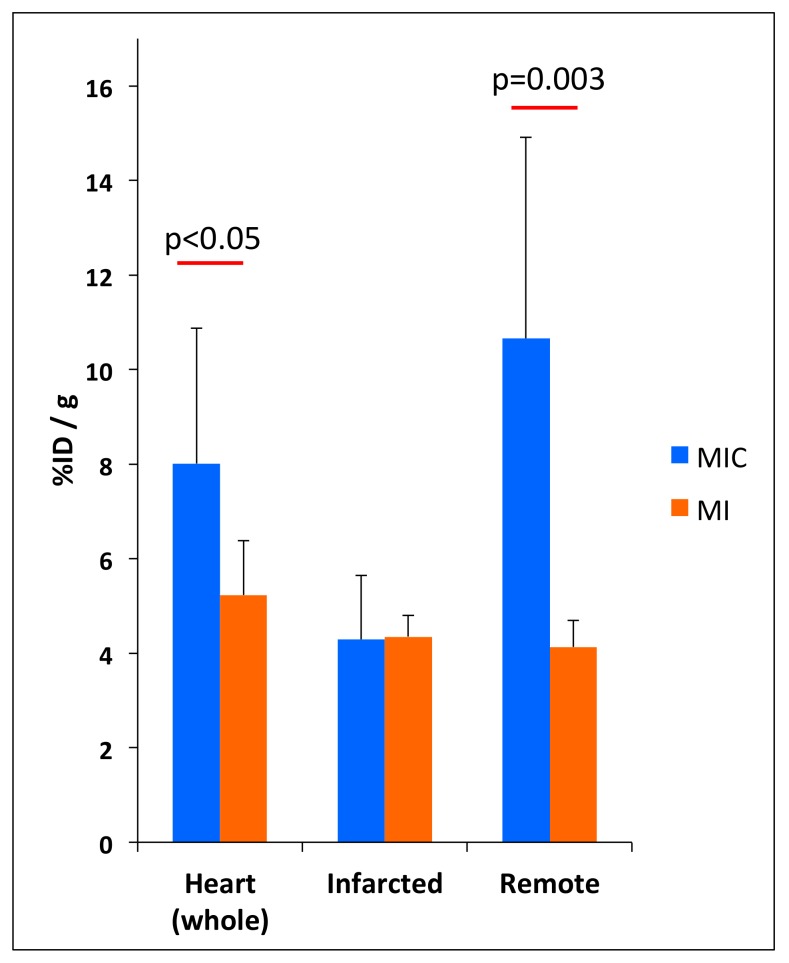
Regional quantitative analysis of 18F-FDG uptake five days after myocardial infarction in the whole heart, infarct, and remote myocardium. Both MI and MIC group were anaesthetized using ketamine/xylazine. Values are presented as mean ± SD. *p*-value was calculated using the student *t*-test.

**Figure 10 cells-08-01613-f010:**
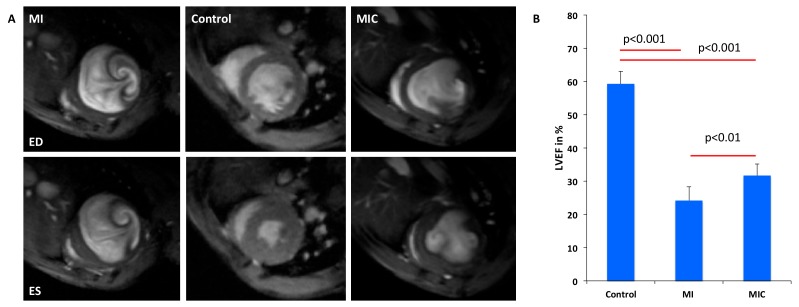
Improved cardiac function after transplantation of embryonic stem cells (ESC) derived CiC following acute myocardial infarction at 3 weeks following MI. (**A**) Magnetic resonance imaging analyses of infarcted animals receiving either CiC (MIC) or matrigel and healthy animals and (**B**) left ventricular ejection fraction was significantly higher in CiC animals vs. control animals (n = 6, *p*-value was calculated using the student *t*-test).

**Figure 11 cells-08-01613-f011:**
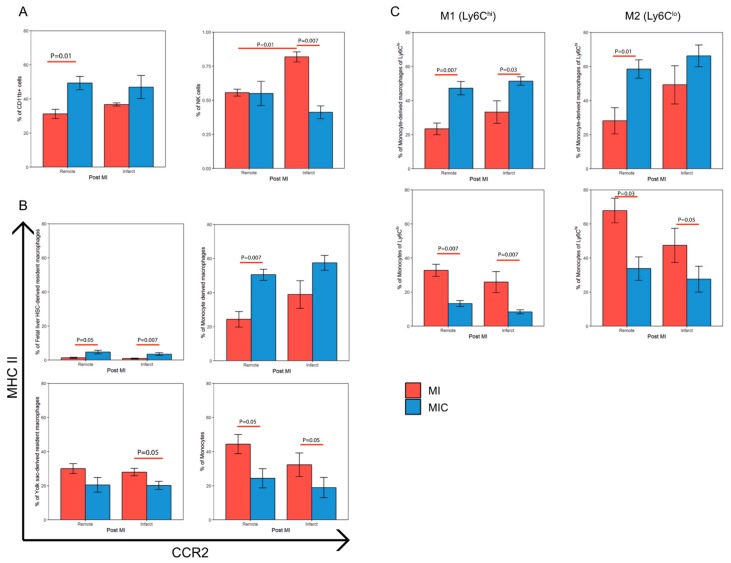
Effect of CiC transplantation on the percentage of various immune cell subpopulations in the heart based on flow cytometric analysis. Mice were subjected to MI/MI+ cells and the remote and infarct area of the hearts were dissected, digested, and the isolated mononuclear cells were stained using various antibodies. These immune cell populations were then characterized using the gating strategy described before and represented as three major groups. (**A**) CD11b^+^ and NK cells, (**B**) populations based on their CCR2/MHC-II expression, and (**C**) contribution of the populations towards the M1 (Ly6C^hi^) and M2 (Ly6C^lo^) cells, n = 5. Values are represented as mean ± SEM. Significance was calculated using the Mann–Whitney test.

**Table 1 cells-08-01613-t001:** Antibodies used for flow cytometry.

Target	Clone	Source
CD45	30-F11	Biolegend
CD11b	M1/70	Biolegend
CD11c	N418	Biolegend
NK1.1	PK136	Biolegend
Ly6G	1A8	Biolegend
Ly6C	Hk1.4	Biolegend
CCR2	475301	R and D
MHC-II	AF6-120.1	Biolegend

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
