# Peer review of "18F-FDG PET-Based Imaging of Myocardial Inflammation Predicts a Functional Outcome Following Transplantation of mESC-Derived Cardiac Induced Cells in a Mouse Model of Myocardial Infarction"

_cells, 2019, doi:10.3390/cells8121613_

Round 1

Reviewer 1 Report

General comments,  the authors have shown a serial investigation of the efficacy of transplantation of cardiac progenitor cells in animals with a myocardial infraction, they demonstrated early inflammatory proc3ss post-MI monitoring by early 18F-FDG PET/CT could be used to predict heart long-term prognostic outcome of stem cells transplantation which is determined by CMR. Actually, the linkage between myocardial restored inflammation and the cardiac outcome has been well described in the pioneering studies. The authors implement this inflammatory metabolism imaging to predict the stem cell therapy outcome is relatively interesting. Additionally, the authors also performed a cell-profile study, aimed to determine the immune-cell distribution and migration in distinct myocardial areas, it indirectly provided evidence of early FDG accumulation dominant by certain immune cell types which is innovative.

Specific comments

The authors need to describe the location of intramyocardial transplantation of CPCs, particular how to do the control in each individual mice.

I’m wondering the rationale the authors performed CMR in separate animal groups, which limited the conclusion.

The authors need to describe more concrete, better showed an example of how did they define/draw the VOIs in the remote area, infarcted area. Which is tricky but could be a significant influencer of the results.

A polar map is preferable to visualize the global FDG retention in the heart.

A series of dynamic imaging is preferable to quantify the blood pool clearance.

Reviewer 2 Report

Vasudevan and co-authors have explored the possibility of using 18F-FDG-PET-based imaging to quantified the effect of transplantation of mESC-derived cells. The study was performed in a mouse model of myocardial infarction (129S6 mouse strain). They found that under, ketamine/xylazine anesthesia,  cell transplantation induced a shift in 18F-FDG-uptake pattern, leading to significantly higher 18F-FDG-uptake in the whole heart as well as the remote area of the heart. This is an interesting point! The possibility of using  18F -FDG-PET for imaging post-infarct inflammation in a mouse model of myocardial infarction is not new. For example, Thackeray et al (article properly cited and discussed by the authors) have shown that 18F-FDG-PET-based imaging (under ketamine/xylazine anesthesia) highlights differences in the 18F-FDG uptake levels between the remote and infarcted myocardium. Differently from Thackeray et al., this paper does not highlight any difference between infarcted and remote areas. This makes me a little bit uncomfortable. I agree with the authors that this incongruity can be due to differences in mouse strains, infarct size, etc... but,  this must be proved. In fact, considering the limited studies in this field, the authors have to first demonstrate their capability in reproducing the “state of the art” (or define a new one) before to apply this methodology in investigating the effect of cell transplantation. In my opinion, additional experiments are required in this respect.

Other points:

1) The manuscript is not well organized. A general reader discovers only during the discussion the reason for the comparison under isoflurane and ketamine/xylazine anesthesia. The introduction section is sufficient to prepare readers for results.

2) Figures must be revised: i) strange symbols are present; ii) The same logic order should be maintained across figures (see for example figure 3 and 4); iii) In figure 7 is impossible to distinguish the two experimental classes; iv) Not clear in figure 3 the three different views (better in figure 4); … etc.

Round 2

Reviewer 1 Report

No further comments.

Reviewer 2 Report

The authors properly addressed all my critical points. Very interesting work.